# Ezrin Regulates the Cell Surface Localization of PD-L1 in HEC-151 Cells

**DOI:** 10.3390/jcm11082226

**Published:** 2022-04-15

**Authors:** Chihiro Tanaka, Takuro Kobori, Rie Okada, Rina Doukuni, Mayuka Tameishi, Yoko Urashima, Takuya Ito, Nobumasa Takagaki, Tokio Obata

**Affiliations:** 1Laboratory of Clinical Pharmaceutics, Faculty of Pharmacy, Osaka Ohtani University, Tondabayashi 584-8540, Japan; u4117078@osaka-ohtani.ac.jp (C.T.); koboritaku@osaka-ohtani.ac.jp (T.K.); u4118025@osaka-ohtani.ac.jp (R.O.); u4118098@osaka-ohtani.ac.jp (R.D.); u4117083@osaka-ohtani.ac.jp (M.T.); urasiyo@osaka-ohtani.ac.jp (Y.U.); 2Laboratory of Natural Medicines, Faculty of Pharmacy, Osaka Ohtani University, Tondabayashi 584-8540, Japan; itoutaku@osaka-ohtani.ac.jp; 3Nobumasa Clinic, Kyoto 601-8041, Japan; nobukin555@gmail.com

**Keywords:** programmed death ligand-1, ezrin, radixin, moesin, endometrioid adenocarcinoma, immune checkpoint inhibitor, cancer immunotherapy

## Abstract

Programmed death ligand-1 (PD-L1) is an immune checkpoint molecule widely expressed on the surface of cancer cells and is an attractive immunotherapeutic target for numerous cancer cell types. However, patients with endometrial cancer derive little clinical benefit from immune checkpoint blockade therapy because of their poor response rate. Despite the increasingly important function of PD-L1 in tumor immunology, the mechanism of PD-L1 localization on endometrial cancer cell surfaces is largely unknown. We demonstrated the contribution of the ezrin, radixin, and moesin (ERM) family, which consists of scaffold proteins that control the cell surface localization of several transmembrane proteins to the localization of PD-L1 on the cell surface of HEC-151, a human uterine endometrial cancer cell line. Confocal immunofluorescence microscopy and immunoprecipitation analysis revealed the colocalization of all the ERM with PD-L1 on the cell surface, as well as their protein–protein interactions. The RNA-interference-mediated knockdown of ezrin, but not radixin and moesin, significantly reduced the cell surface expression of PD-L1, as measured by flow cytometry, with little impact on the PD-L1 mRNA expression. In conclusion, among the three ERM proteins present in HEC-151 cells, ezrin may execute the scaffold function for PD-L1 and may be mainly responsible for the cell surface localization of PD-L1, presumably via the post-translational modification process.

## 1. Introduction

Endometrial cancer is the most common cancer in the female genitalia, with 417,306 new cases worldwide in 2020 [1]. Endometrioid adenocarcinomas represent 80% of endometrial carcinomas [2]. The number of cases has been increasing rapidly in recent years and is expected to double by 2040 [3]. Surgery, radiation therapy, and chemotherapy are the first-line treatment modalities available for endometrioid adenocarcinomas. However, treatment options are limited for recurrent and metastatic cases, with an extremely low 5-year survival rate in women with advanced or recurrent disease [4].

Recently, immune checkpoint blockade (ICB) therapy has been increasingly recognized as an innovative treatment for cancer [5,6,7]. Programmed death ligand-1 (PD-L1), an immune checkpoint molecule, is distributed in numerous cells, such as dendritic and epidermal cells, and interacts with programmed death-1 (PD-1) expressed on T cells to inactivate T cells and suppress excessive immune responses [8,9]. PD-L1 is also widely present in cancer cells and is involved in the evasion of immunity by cancer cells [8,10]. Immune checkpoint inhibitors targeting immune checkpoint molecules such as PD-L1 promote T cell activation and enhance cancer immunity by inhibiting the interaction between immune checkpoint molecules [11]. The efficacy of ICB-based therapies has been widely reported in numerous cancer cell types [7,11]. Notably, pembrolizumab, an anti-PD-1 antibody (Ab), has recently provided survival benefits in patients with endometrial cancer, leading to its approval as an ICB therapy [12,13]. However, the response rate is insufficient [12,13]; thus, further exploration of therapeutic targets is warranted.

Accumulating evidence indicates a complex regulatory mechanism of PD-L1 expression involving diverse cellular events [14,15,16]. Since PD-L1 is a plasma membrane protein, several post-translational modification factors for PD-L1 have received much attention in recent years [14,15,16,17,18]. The family members of the ezrin, radixin, and moesin (ERM) protein execute scaffold functions for the cell surface localization of several transmembrane proteins, such as some drug transporters involved in multidrug resistance, epidermal growth factor receptor (EGFR) 2, and cluster of differentiation (CD) 20 [19,20,21,22,23,24,25]. Intriguingly, we recently reported that in a small number of human cancer cell types, the ERM family proteins modulate the cell surface localization of PD-L1 in a different way by serving as scaffold proteins. However, it is unclear whether the ERM family executes the scaffold function to stabilize PD-L1 in the cell surfaces of uterine endometrial cancer cells.

Here, we demonstrated the contribution of ERM family proteins to the cell surface localization of PD-L1 in HEC-151 cells, a human uterine endometrial adenocarcinoma cell line.

## 2. Materials and Methods

### 2.1. Cell Culture

The human uterine endometrial cancer cell line HEC-151, established by Kuramoto H. et al. [26], was obtained from the Japanese Collection of Research Bioresources Cell Bank (JCRB1122; Osaka, Japan). HEC-151 cells were cultured in Dulbecco’s Modified Eagle’s Medium (DMEM), which contained 1500 mg/L glucose (FUJIFILM Wako Pure Chemical, Osaka, Japan) supplemented with 10% heat-inactivated fetal bovine serum (FBS) (BioWest, Nuaillé, France). During cultivation, cells were maintained at 37 °C in a humidified atmosphere with 5% CO_2_ and cultured up to 70–80% of flasks.

### 2.2. siRNA Treatment

HEC-151 cells were seeded at a density of 2.0 × 10^4^ cells/well in 24-well cell culture plates and incubated overnight to allow cell attachment. Small interfering RNAs (siRNAs) targeting each human gene and nontargeting control siRNA were introduced into cells at a concentration of 5 nM using the Lipofectamine RNAiMAX at doses of 0.10 µL/1.0 × 10^4^ cells. After siRNA and the transfection reagent were added, the cells were cultured continuously for 4 days without replacing the medium. All the reagents used for siRNA treatment were purchased from Thermo Fisher Scientific (Tokyo, Japan).

### 2.3. Real-Time Reverse Transcription–Polymerase Chain Reaction

Total RNA was extracted from HEC-151 cells using an ISOSPIN Cell and Tissue RNA kit (Nippon Gene, Tokyo, Japan) according to the manufacturer’s protocol. The total RNA concentration and purity were evaluated using a NanoDrop Lite spectrophotometer (Thermo Fisher Scientific). Subsequently, real-time reverse transcription (RT)–polymerase chain reaction (PCR), followed by the calculation for relative quantification of each transcript, was performed as described previously [27,28]. The sequences of gene-specific PCR primers are listed in Appendix A. All the reagents and equipment used for the real-time RT-PCR reaction were obtained from TaKaRa Bio (Shiga, Japan) and Bio-Rad Laboratories (Hercules, CA, USA), respectively.

### 2.4. Confocal Laser Scanning Microscopy Analysis

Confocal laser scanning microscopy (CLSM) analysis was performed as described previously, with some modifications [27,29]. Single and double immunofluorescence staining were performed to determine the intracellular localization of ezrin, radixin, moesin, and PD-L1, or the colocalization of PD-L1 with ezrin, radixin, and moesin, respectively.

#### 2.4.1. Single Immunofluorescence Staining

HEC-151 cells seeded at 2.0 × 10^5^ cells on a polylysine-coated, 35 mm glass-bottom dish (Matsunami Glass, Osaka, Japan) were incubated overnight to allow their attachment. The cells were fixed with 4% paraformaldehyde and then permeabilized with 0.5% Triton-X100, subsequently incubated in a blocking buffer containing Dulbecco’s phosphate-buffered saline (D-PBS), which was supplemented with 0.3 M glycine, 10% normal goat serum, 1% bovine serum albumin, and 0.1% Tween-20. After that, the cells were incubated overnight at 4 °C with the respective primary Abs and then incubated with Alexa Fluor 488-conjugated secondary Ab for ezrin, radixin, and moesin. Subsequently, the plasma membrane was counterstained with a tetramethylrhodamine (TRITC)-conjugated phalloidin, which is a high-affinity F-actin probe. The preserved cells were observed and photographed at 0.5–1.5 µm intervals on the *z*-axis at an original magnification of ×20 with a Nikon Al confocal laser microscope system and NIS-Elements Ar Analysis software (Nikon Instrument, Tokyo, Japan). All the Abs used in this study are listed in Appendix A.

#### 2.4.2. Double Immunofluorescence Staining

Cell fixation, permeation, and blocking before the Ab reaction were conducted in the same way as described above. Then, the cells were incubated overnight at 4 °C with the respective primary Abs against each ERM and then incubated with Alexa Fluor 594-conjugated secondary Ab. Thereafter, the cells were incubated overnight at 4 °C with an Alexa Fluor 488-conjugated anti-PD-L1 Ab. The subsequent procedure was conducted as described in the section on single immunofluorescence staining.

### 2.5. Western Blotting

HEC-151 cells were lysed in radioimmunoprecipitation assay (RIPA) buffer containing a protease inhibitor cocktail on ice. Lysates containing equal amounts of proteins were heated at 97 °C for 5 min in 2× sample buffer consisting of 0.125 M Tris-HCl, 4% sodium dodecyl sulfate (SDS), 20% glycerin, 0.01% bromophenol blue, and 10% 2-mercaptoethanol. Subsequently, the proteins were separated by SDS-polyacrylamide gel electrophoresis (PAGE) and transferred to a nitrocellulose membrane. The membrane was blocked with 5% skim milk in PBS-T and then probed overnight with the respective primary Abs at 4 °C. The membranes were incubated with the respective HRP-conjugated secondary Abs and visualized with an enhanced chemiluminescence system on a LuminoGraphII EM (ATTO, Tokyo, Japan). All the original source images for immunoblots are given in Appendix A.

### 2.6. Immunoprecipitation

Immunoprecipitation assays were conducted as previously described [27,28,29,30], with some modifications. Briefly, 500 μL of whole-cell lysates collected in RIPA buffer containing a protease inhibitor cocktail was filtered through 50 μL of protein A beads (nProtein A Sepharose 4 Fast Flow; Cytiva, Tokyo, Japan) for 1 h at 4 °C on a rotating wheel to eliminate non-specific interactions. The pre-cleaned lysates were incubated on a rotating wheel at 4 °C overnight with an anti-PD-L1 Ab or control IgG Ab (1:30) and mixed with 50 μL of protein A beads on a rotating wheel at 4 °C for 3 h. After incubation, protein A beads were collected and rinsed three times with RIPA buffer, followed by heating at 97 °C for 5 min in 2× sample buffer (Nacalai Tesque) before immunoblotting.

### 2.7. Flow Cytometric Assay

Flow cytometry analysis was performed as previously described [27,28,29,30], with some modifications. Single-cell suspensions were incubated with an allophycocyanin (APC)-conjugated anti-PD-L1 Ab (4.0 μg/tube) in a labeling buffer consisting of D-PBS, 5% normal horse serum, and 1% sodium azide at 4 °C for 1 h. Thereafter, the mean fluorescence intensity of APC-PD-L1 on the HEC-151 cell surfaces was analyzed using a Cell Analyzer EC800 and EC800 Analysis software (Sony Imaging Products and Solutions, Tokyo, Japan).

### 2.8. Statistical Analysis

Data are expressed as mean ± standard error of the mean (SEM). Statistical analysis was conducted using Prism version 3 (GraphPad Software, La Jolla, CA, USA). Statistical significance was assessed using one-way analysis of variance (ANOVA), followed by multiple comparisons using Dunnett’s test. Differences were considered statistically significant at *p* < 0.05.

## 3. Results

### 3.1. Expression Profiles of PD-L1 and Each ERM at mRNA and Protein Levels in HEC-151 Cells

We thoroughly evaluated the expression patterns of ERM and PD-L1 at mRNA levels in numerous human endometrial cancer cell lines by utilizing the database of the Cancer Dependency Map (DepMap) portal [31,32,33]. The relative expression level of PD-L1 mRNA in HEC-151 cells was higher than those in the other cells. Furthermore, HEC-151 cells have abundant levels of ezrin, radixin, and moesin (Figure 1a). As shown in the amplification curve obtained using real-time RT-PCR, ERM and PD-L1 were expressed at mRNA levels in sufficient amounts (Figure 1b). In addition, Western blotting analysis indicated the existence of ERM proteins and PD-L1 at the protein level (Figure 1c). These results implied sufficient mRNA and protein expressions of ERM proteins and PD-L1 in HEC-151 cells.

### 3.2. Plasma Membrane Localization of PD-L1 and ERM in HEC-151 Cells

Intracellular localization of ERM and PD-L1 in HEC-151 cells was confirmed by immunofluorescence confocal laser scanning microscopy. In HEC-151 cells, all three ERM and PD-L1 were primarily detected in the cell surface region labeled with F-actin (Figure 2a–d). Furthermore, the results of immunofluorescence double staining indicated that the fluorescence signals of all three ERM were highly overlapped with those of PD-L1 (Figure 3a–c). The results of confocal laser scanning microscopy indicated that ERM and PD-L1 were colocalized on the surfaces of HEC-151 cells. This is the first report to demonstrate the intracellular colocalization of PD-L1 with ERM in human endometrioid adenocarcinoma cells.

### 3.3. Protein–Protein Interaction between PD-L1 and ERM in HEC-151 Cells

We analyzed the protein–protein interaction between PD-L1 and ERM in HEC-151 cells. The expression of PD-L1 and all three ERM was observed in the immune precipitates of HEC-151 cells that were pulled down using an Ab against PD-L1 (Figure 4). In contrast, they were not detected in samples that were pulled down using a control IgG Ab. The results of the immunoprecipitation assay demonstrated for the first time that PD-L1 endogenously interacts with all three ERM in HEC-151 cells.

### 3.4. Effect of siRNAs against ERM on the Expression Levels of PD-L1 in HEC-151 Cells

We checked the effects of ERM siRNA on the expression of PD-L1 at the mRNA level. siRNAs against ezrin, radixin, and moesin strongly decreased each target mRNA expression level (Appendix A). Treatment with moesin siRNA significantly upregulated PD-L1 mRNA expression. However, no alterations in the mRNA expression of PD-L1 were observed in the cells treated with ezrin and radixin siRNAs, even though siRNAs against PD-L1 strongly decreased the PD-L1 mRNA expression (Figure 5a). Finally, we examined the effects of ERM knockdown on PD-L1 expression in the cell surface using flow cytometry. The results showed that the cell surface expression of PD-L1 was significantly reduced by the gene silencing of ezrin but not by that of radixin and moesin (Figure 5b,c). Collectively, ezrin may execute a scaffold function to stabilize PD-L1 on the surfaces of HEC-151 cells with little impact on the transcriptional process of PD-L1.

## 4. Discussion

In the present study, we validated the detectability of all three ERM at mRNA and protein levels in HEC-151 cells. Other studies have also indicated that ezrin and moesin are highly expressed at mRNA and/or protein levels in human endometrioid carcinoma tissues and several human endometrial epithelial cancer cell lines, such as RL-95, AN3CA, Ishikawa, HEC-50, and HEC-1-A [34,35,36,37,38,39,40]. In contrast, no studies have examined the expression of radixin in endometrioid cancers. Although the expression patterns of ERM depend on the types of human endometrioid cancer cells, as shown in Figure 1a, HEC-151 cells contain sufficient levels of all three ERM proteins to evaluate the role of each ERM in the cell surface localization of PD-L1. We also confirmed the PD-L1 expressions at the mRNA and protein levels in HEC-151 cells, the results of which were in line with previous findings which showed that PD-L1 is highly expressed in numerous human endometrioid cancer cell lines, including Ishikawa, HEC-50, HEC-1-A, HOUA-I, and RL95-2 [41,42,43], as well as in clinical uterine corpus endometrial carcinoma [44,45]. Furthermore, our CLSM data showed the colocalization of PD-L1 with ERM in the cell surface regions of HEC-151 cells due to the predominant subcellular distribution of these proteins on the cell surfaces. Altogether, our present study demonstrated that in HEC-151 cells, ERM proteins and PD-L1 are expressed at sufficient levels and are highly colocalized in the cell surface regions.

PD-L1 expression is regulated by a number of intracellular events. Recently, much attention has been paid to the post-translational modification process of PD-L1 to determine its cell surface localization [15,16,46]. As some of the key players in this mode, ERM proteins anchor numerous transmembrane proteins to the actin cytoskeleton, contributing to their cell surface localization. Interestingly, we recently found that among ERM proteins, ezrin modulates the cell surface localization of PD-L1 in human uterine cervical adenocarcinoma (HeLa), choriocarcinoma (JEG-3), and colon adenocarcinoma (LS180) cells, in which the expression levels of ezrin are higher than those of radixin and moesin, based on comprehensive gene expression analysis using DepMap and our RT-PCR experiments [28,29,30]. We also demonstrated that as a predominant ERM protein, radixin primarily regulates the cell surface localization of PD-L1 in human pancreatic ductal adenocarcinoma cells (KP-2) [27]. On the other hand, Meng et al. elucidated that in human breast cancer adenocarcinoma, moesin protects PD-L1 against the proteasomal degradation system, leading to the stabilization of the plasma membrane PD-L1, although the functions of ezrin and radixin have yet to be examined [47]. Likewise, the results of immunoprecipitation showed the intrinsic interaction of PD-L1 with ezrin, radixin, and moesin in HEC-151 cells, as is the case in the results obtained from HeLa, JEG-3, LS180, and KP-2 cells [27,28,29,30]. Interestingly, the RNA-mediated interference of ezrin, but not radixin and moesin, considerably suppressed the cell surface expression of PD-L1 without altering its mRNA levels, implying that among ERM proteins, ezrin predominantly modulates the cell surface localization of PD-L1 in HEC-151 cells. The ERM proteins that predominantly modulate the cell surface expression of PD-L1 may be determined by the expression pattern of ERM, depending on the cancer cell type. Taken together, ezrin principally regulates the cell surface expression of PD-L1 in HEC-151 cells, as shown in our previous findings [28,29,30].

Unpredictably, the RNA-mediated interference of moesin dramatically upregulated the PD-L1 mRNA expression in HEC-151 cells. One possibility is that the inhibition of moesin might produce the major inducers for PD-L1 mRNA, including interferon (IFN)-γ, tumor necrosis factor (TNF)-α, and interleukin (IL)-6 [14,48,49]. In fact, knockdown of moesin greatly increases the TNF and IL-6 mRNA expressions in LS180 cells, leading to an increase in PD-L1 mRNA expression [29]. Additionally, others have reported that a neutralizing Ab against moesin induced IFN-γ, TNF-α, and IL-6 release from several human immune cells [50,51]. These previous observations might partly support the present finding that the gene silencing of moesin induces the PD-L1 mRNA expression in HEC-151 cells. These complex questions should be addressed in future studies.

In summary, among the three ERM proteins present in HEC-151 cells, ezrin may be primarily responsible for scaffolding protein for PD-L1. Therefore, new drug modalities targeting ezrin may be effective for reducing the PD-L1 expression on the cell surface of human endometrial adenocarcinoma and providing a novel therapeutic option to enhance the response rate of ICB therapies. In vivo experiments using xenograft model mice should be addressed in our future studies for potential translation into clinical applications.

## Figures and Tables

**Figure 1 jcm-11-02226-f001:**
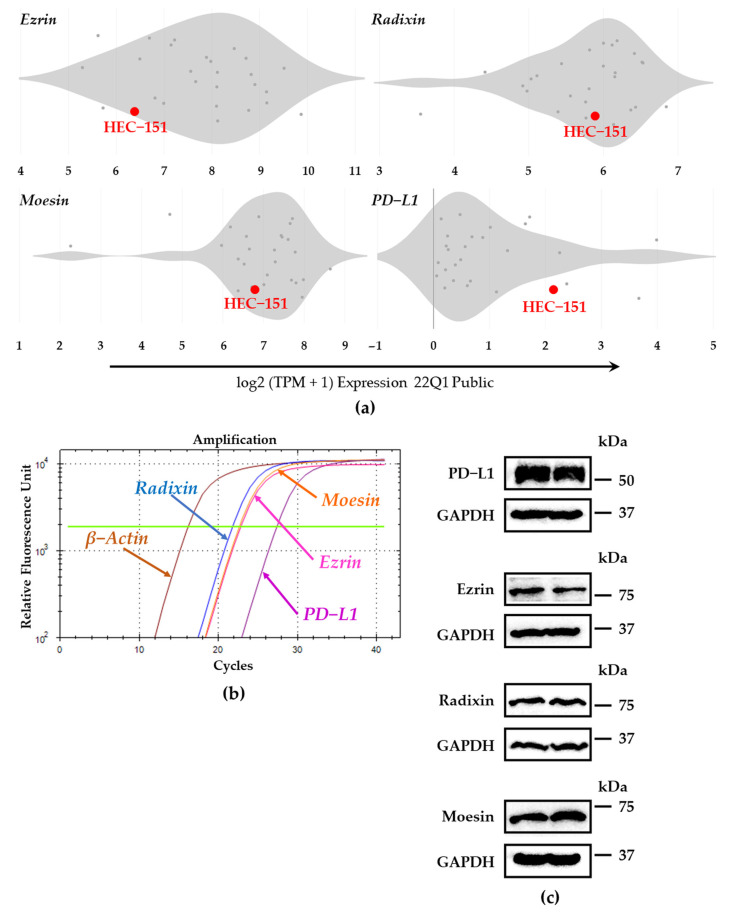
Expression profiles of ezrin, radixin, moesin (ERM), and programmed death ligand-1 (PD -L1) at mRNA and protein levels in HEC-151 cells. (**a**) Violin plots showing the median gene expression (log2 (TPM + 1)) of ERM and PD-L1 in numerous human endometrial cancer cells obtained from the Cancer Dependency Map (DepMap) portal data explorer, DepMap 22Q1 Public. (**b**) Amplification curves for each target gene in HEC-151 cells, as measured by real-time reverse transcription–polymerase chain reaction. (**c**) Immunoblot images of each protein in HEC-151 cells. Upper panels; ezrin, radixin, moesin, and PD-L1, lower panels; corresponding glyceraldehyde-3-phosphate dehydrogenase (GAPDH) shown in duplicate. Molecular weights; kDa.

**Figure 2 jcm-11-02226-f002:**
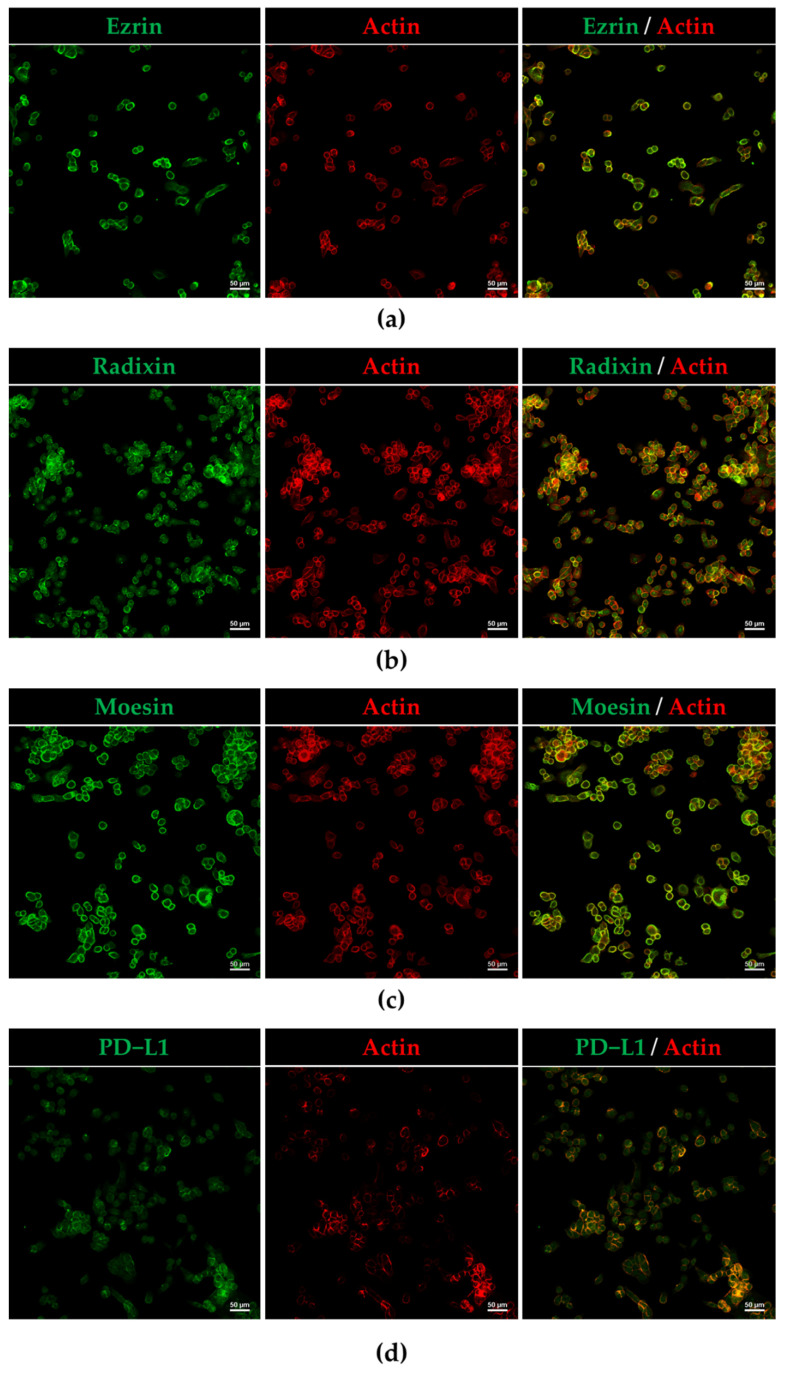
Plasma membrane localization of ezrin, radixin, moesin (ERM), and programmed death ligand-1 (PD-L1) in HEC-151 cells. Three-dimensional reconstructions of optically sectioned HEC-151 cells as analyzed by confocal laser scanning microscopy showed the intracellular localization of (**a**) ezrin, (**b**) radixin, (**c**) moesin, and (**d**) PD-L1 (Alexa Fluor 488) counterstained with actin (tetramethylrhodamine; TRITC), a typical plasma membrane marker. Scale bars: 50 μm.

**Figure 3 jcm-11-02226-f003:**
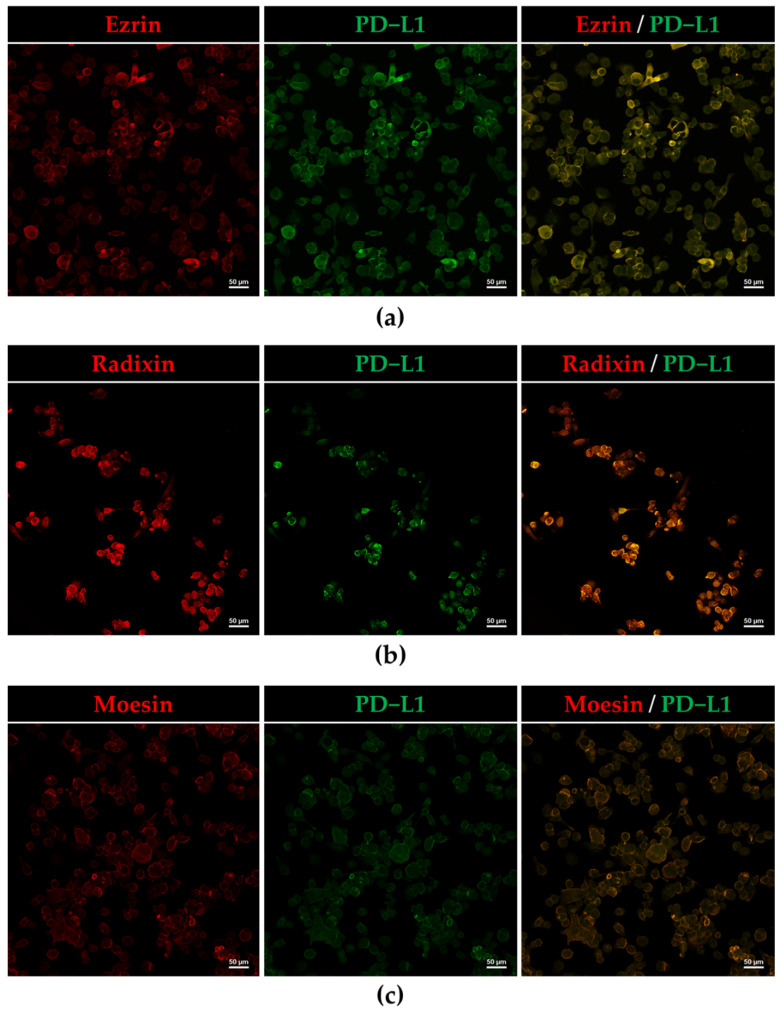
Programmed death ligand-1 (PD-L1) was colocalized with ezrin, radixin, and moesin in the plasma membrane of HEC-151 cells. Colocalization of (**a**) ezrin, (**b**) radixin, and (**c**) moesin (Alexa Fluor 594) with PD-L1 (Alexa Fluor 488) in the plasma membrane as analyzed by confocal laser scanning microscopy. Scale bars: 50 μm.

**Figure 4 jcm-11-02226-f004:**
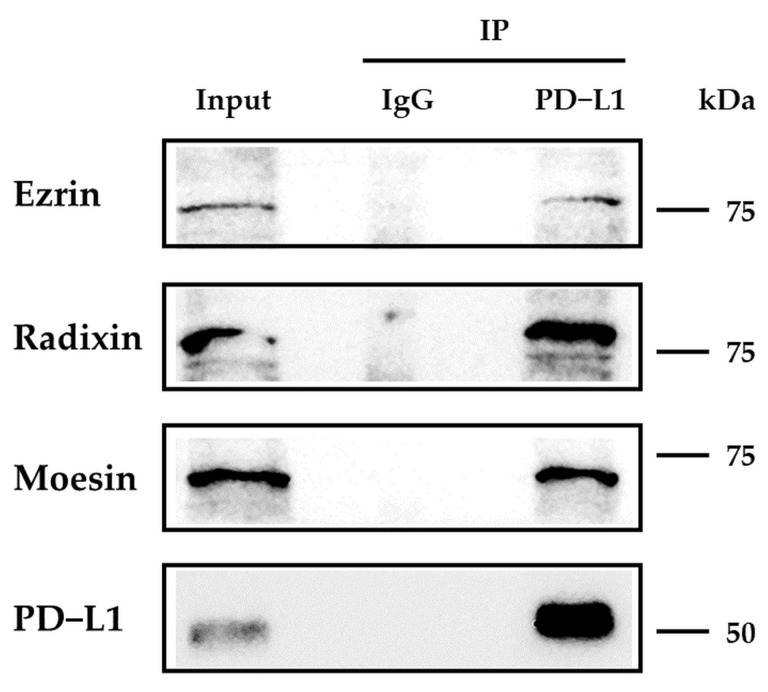
Immunoblots for programmed death ligand-1 (PD-L1) co-immunoprecipitation in HEC-151 cells. Immunoreactive bands of ezrin, radixin, moesin, and PD-L1 in the input and those co-immunoprecipitated (IP) with a control IgG or an anti-PD-L1 Ab. Molecular weights; kDa.

**Figure 5 jcm-11-02226-f005:**
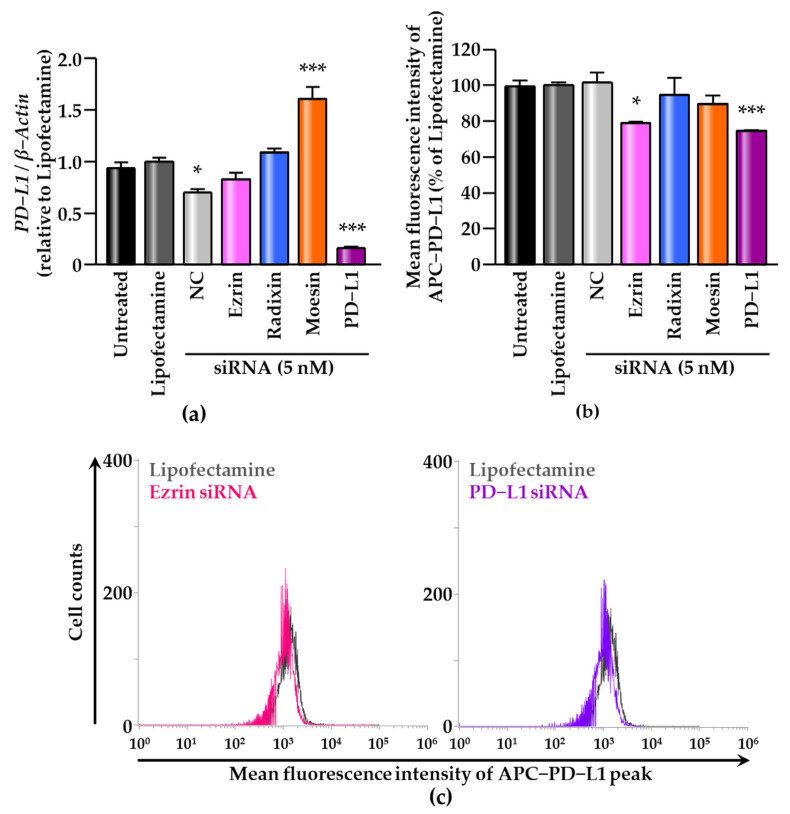
Effects of RNA-mediated interference of ezrin, radixin, and moesin on the expression of programmed death ligand-1 (PD-L1) in HEC-151 cells. (**a**) Relative mRNA expression of PD-L1 normalized with β-Actin was determined via a reverse transcription–polymerase chain reaction. *n* = 3, *** *p* < 0.001, * *p* < 0.05 vs. Lipofectamine. (**b**) The mean fluorescence intensities of allophycocyanin (APC)-labeled PD-L1 relative to Lipofectamine alone; *n* = 3, *** *p* < 0.001, * *p* < 0.05 vs. Lipofectamine. (**c**) Histograms for APC-labeled PD-L1 fluorescence in the surface of HEC-151 cells treated with Lipofectamine (gray), ezrin siRNA (red), and PD-L1 siRNA (purple), as measured by flow cytometry. All data were expressed as the mean ± SEM and analyzed by one-way ANOVA followed by Dunnett’s test. Nontargeting control; NC.

## Data Availability

The datasets used and analyzed in this study are available from Cancer Cell Line Encyclopedia (https://sites.broadinstitute.org/ccle/datasets, accessed on 7 March 2022), DepMap, Broad (2022): DepMap 22Q1 Public. figshare. Dataset (https://doi.org/10.6084/m9.figshare.19139906.v1, accessed on 7 March 2022). Other data are contained within the article and Appendix A.

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
