# Peer review of "Ezrin Regulates the Cell Surface Localization of PD-L1 in HEC-151 Cells"

_jcm, 2022, doi:10.3390/jcm11082226_

Round 1
Reviewer 1 Report
Dear authors
Please send the raw data of western blot and IP.
This paper highlights the regulation of PD-L1 in localization mainly by Erzin and radixin and moesin in endometrial adenocarcinoma cells. This paper holds a lot of strength in the novelty of topics discussed esp, the unknown role of ERM in stabilization of PD-L1 in the surface of endometrial cancer cells. Furthermore, the authors provide good depth of discussion, considering the number of relevant papers mentioned and the way they were compared to this paper. This paper has a potential to be accepted, but western blot and IP images are unclear. That's why I asked for raw data of both.
The authors better implement some xenograft in nude mice to evaluate the in-vivo condition. We can check elaborately if the PD-L1 modification that this paper implemented can really make a difference.
In summary, this paper is trying to make progress to address the poor response rate of PD-L1 in endometrial cancer.
regards,
Author Response
Response to Reviewer 1’s Comments
We would like to thank #Reviewer 1 for the greatest evaluation on our manuscript. We have carefully read your comments and suggestions and have made the corrections in the revised version of manuscript. Detailed responses to your comments are listed below, and we highlighted all changes with word track changes in the file labeled ‘Revised Manuscript with Track Changes’. We hope this revised manuscript would be satisfactory for publication in Journal of Clinical Medicine.
Comment 1. Please send the raw data of western blot and IP.
Reply Comments.
We would like to appreciate #Reviewer 1’s valuable suggestion. According to #Reviewer 1’s comment, we have added following data and sentence in the Materials and Methods and Supplementary Materials.
Materials and Methods (Line 131 - 132)
All the original source images for immunoblots are given in Figure S1.
Supplementary Materials (Line 24 – 28)
Original source images for immunoblots.
Figure S1. Original source images for immunoblots. The original western blotting membrane to detect the protein expression of programmed death ligand-1 (PD-L1), ezrin, radixin, and moesin as well as the corresponding glyceraldehyde-3-phosphate dehydrogenase (GAPDH) used as a loading control shown in Figure 1c or Figure 4.
Comment 2. The authors better implement some xenograft in nude mice to evaluate the in-vivo condition. We can check elaborately if the PD-L1 modification that this paper implemented can really make a difference.
In summary, this paper is trying to make progress to address the poor response rate of PD-L1 in endometrial cancer.
Reply Comments.
We would like to appreciate #Reviewer 1’s valuable and meaningful suggestion. As #Reviewer 1 pointed out, we should address in vivo experiments that aims at clinical application of our idea in our future studies. According to #Reviewer 1’s suggestion, we have incorporated following sentence into the Results and Discussion sections.
Thank you again for your constructive suggestion.
Discussion (Line 299 - 301)
In order to lead our idea for potential translation into clinical applications, the in vivo experiments using a xenograft model mice should be addressed in our future studies.
Reviewer 2 Report
Authors conducted a fruitful study which proposed demonstrate the contribution of ERM family proteins to cell surface local- 66 ization of PD-L1 in HEC-151 cells, a human uterine endometrial cancer cell line. The proposed hypothesis would be valuable. In this manner, it is a significant and adequate paper.
Searching the literature, one finds that researchers had previously planned similar work on different cell lines in this area.This does not make the study significant and unique but performing the experiment on endometrial cancer cell lines; makes the study challenging in this area.
The methodology of the paper is quite common. The laboratory techniques used, the experimental methods and the presentation of the results are quite ideal and sufficient. On the other hand, chosen keywords of the whole research is well defined.
While the proposed framework seems novel and idea driven in the paper is promising, authors should emphasize the finding and their future plan ( for instance ; switching to animal experiments now instead of cell lines) more and more in the results section, if they are so.
The implications and phases of the conducted framework are designed well. also, nearly all the applications are explained clearly. All parts such as results and included parts are defined well. the reasons for excluding those should be explained more. The theoretical implications are defined well.
The communication quality of the paper is clear. Authors express what they do well. Overall, the proposed hypothesis is promising and well testing. The language of the paper is well and without any typo errors, as far as I could detected.
Author Response
Response to Reviewer 2’s Comments
We would like to thank #Reviewer 2 for the greatest evaluation on our manuscript. We have carefully read your comments and suggestions and have made the corrections in the revised version of manuscript. Detailed responses to your comments are listed below, and we highlighted all changes with word track changes in the file labeled ‘Revised Manuscript with Track Changes’. We hope this revised manuscript would be satisfactory for publication in Journal of Clinical Medicine.
Comment 1. While the proposed framework seems novel and idea driven in the paper is promising, authors should emphasize the finding and their future plan (for instance; switching to animal experiments now instead of cell lines) more and more in the results section, if they are so.
Reply Comments.
We would like to thank #Reviewer 2 for the greatest evaluation on our manuscript. According to #Reviewer 2’s meaningful and constructive suggestion, we have incorporated following sentence into the Results and Discussion sections.
Results (Line 187 - 190)
This is the first report to demonstrate the intracellular colocalization of PD-L1 with ERM in the human endometrioid adenocarcinoma cells.
Results (Line 208 - 210)
The results of immunoprecipitation assay demonstrated for the first time that PD-L1 endogenously interacts with all the three ERM proteins in HEC-151 cells.
Discussion (Line 299 - 301)
In order to lead our idea for potential translation into clinical applications, the in vivo experiments using a xenograft model mice should be addressed in our future studies.